# Synthesis of Magnetic Iron Oxide-Incorporated Cellulose Composite Particles: An Investigation on Antioxidant Properties and Drug Delivery Applications

**DOI:** 10.3390/pharmaceutics15030732

**Published:** 2023-02-22

**Authors:** Arifa Naznin, Palash Kumar Dhar, Sagar Kumar Dutta, Sumon Chakrabarty, Utpal Kumar Karmakar, Pritam Kundu, Muhammad Sarwar Hossain, Hasi Rani Barai, Md. Rezaul Haque

**Affiliations:** 1Chemistry Discipline, Khulna University, Khulna 9208, Bangladesh; 2Pharmacy Discipline, Khulna University, Khulna 9208, Bangladesh; 3Department of Chemistry, Sogang University, Seoul 04107, Republic of Korea; 4Department of Mechanical Engineering, Yeungnam University, Gyeongsan 38541, Republic of Korea

**Keywords:** sugarcane bagasse, waste tissue paper, nanocomposites, antioxidant properties, drug delivery

## Abstract

In recent years, polymer-supported magnetic iron oxide nanoparticles (MIO-NPs) have gained a lot of attention in biomedical and healthcare applications due to their unique magnetic properties, low toxicity, cost-effectiveness, biocompatibility, and biodegradability. In this study, waste tissue papers (WTP) and sugarcane bagasse (SCB) were utilized to prepare magnetic iron oxide (MIO)-incorporated WTP/MIO and SCB/MIO nanocomposite particles (NCPs) based on in situ co-precipitation methods, and they were characterized using advanced spectroscopic techniques. In addition, their anti-oxidant and drug-delivery properties were investigated. Field emission scanning electron microscopy (FESEM) and X-ray diffraction (XRD) analyses revealed that the shapes of the MIO-NPs, SCB/MIO-NCPs, and WTP/MIO-NCPs were agglomerated and irregularly spherical with a crystallite size of 12.38 nm, 10.85 nm, and 11.47 nm, respectively. Vibrational sample magnetometry (VSM) analysis showed that both the NPs and the NCPs were paramagnetic. The free radical scavenging assay ascertained that the WTP/MIO-NCPs, SCB/MIO-NCPs, and MIO-NPs exhibited almost negligible antioxidant activity in comparison to ascorbic acid. The swelling capacities of the SCB/MIO-NCPs and WTP/MIO-NCPs were 155.0% and 159.5%, respectively, which were much higher than the swelling efficiencies of cellulose-SCB (58.3%) and cellulose-WTP (61.6%). The order of metronidazole drug loading after 3 days was: cellulose-SCB < cellulose-WTP < MIO-NPs < SCB/MIO-NCPs < WTP/MIO-NCPs, whereas the sequence of the drug-releasing rate after 240 min was: WTP/MIO-NCPs < SCB/MIO-NCPs < MIO-NPs < cellulose-WTP < cellulose-SCB. Overall, the results of this study showed that the incorporation of MIO-NPs in the cellulose matrix increased the swelling capacity, drug-loading capacity, and drug-releasing time. Therefore, cellulose/MIO-NCPs obtained from waste materials such as SCB and WTP can be used as a potential vehicle for medical applications, especially in a metronidazole drug delivery system.

## 1. Introduction

Cellulose is considered the most abundant and cost-effective naturally occurring biopolymer, and it is widely used as a precursor for the production of eco-friendly, biocompatible, and high-performance functional materials [1,2,3]. Several studies have explored the possibility that cellulose could be utilized as an excellent polymeric support due to its outstanding mechanical characteristics, large surface area, and almost inexhaustible, renewable, and biodegradable properties [1,4,5]. Moreover, micro- or nanocellulose could be used in food engineering, polypeptides, vaccines, genes, catalysis, nucleic acids, proteins, cosmetics, water purification, construction, electronics, controlled drug delivery, and other fields [6,7,8,9,10].

SCB is a plentiful fibrous waste that is produced on a large scale (540 million metric tons per year globally) [3]. It is normally used for animal feed, enzymes, paper, and biofuel conversion applications. Owing to its high content of cellulose (40–50%), SCB is a promising source of cellulose fiber [3]. Similarly, WTP could be a potential source of raw material for the production of cellulose particles. Every year, several million tons of WTP are produced and used globally, which undoubtedly gives rise to a tremendous amount of waste materials. These are typically burned or discarded in open places, which may cause several health implications and environmental problems. Hence, the proper utilization and conversion of WTP and SCB into value-added materials such as cellulose could be a promising way of recycling those wastes.

Nowadays, the synthesis of nanomaterials using natural or biological sources is gaining momentum owing to their prospective applications in drug delivery, catalysis, sensors, biomedical devices, water treatment, etc. [2,7,8,9,10,11]. Consequently, researchers are focusing their attention on designing metal-incorporated bio-composites that can minimize the use and generation of hazardous substances [10]. In particular, micro- or nanocrystalline cellulose particles possess a highly porous structure with several hydroxyl groups; so, they could be a good platform for anchoring metallic particles [1,2,12]. In this regard, it is expected that the development of metal-incorporated cellulose composite particles would help to produce cost-effective, durable, and almost inexhaustible systems suitable for numerous applications in health and hygiene products, drug delivery, wound dressing, medical equipment coating, and textile materials [8,13,14].

Among the different types of nanomaterials, MIO-NPs, and MIO-incorporated NCPs have drawn much attention in drug delivery due to their facile fabrication and surface functionalization, reusability, low toxicity, biocompatibility, and biodegradability [1,15]. In addition, the excellent magnetic properties of MIO-NP_S_ endow them with extra proficiency over the other drug delivery systems, as represented by the therapeutic (e.g., magnetic targeting and hyperthermia) and diagnostic proficiencies (e.g., as a contrast agent in magnetic resonance imaging) [16,17]. Furthermore, due to the flexible surface chemistry of MIO-NPs, many drugs and bioactive molecules can be bound or loaded onto the surface, forming a drug-conjugated system [18]. In addition, the unusual properties of many drugs, such as poor solubility, nonspecific delivery, and short circulating half-lives, can be overwhelmed by the incorporation of MIO-NPs [16,17]. These exclusive characteristics can be exploited in the development of nanomaterials suitable for medical and drug delivery applications. Some of the relevant studies in this field are the synthesis of jute cellulose/MIO-NCPs and cellulose/MIO/Ag-NCPs by a green bio-reduction method for the evaluation of antioxidant properties [13]; the generation of magnetic NCPs for pharmaceutical and drug delivery applications [19]; the preparation of cellulose/MIO-NCPs from cotton waste via a green route and the assessment of their swelling rate, drug-loading, and drug-releasing behavior in a metronidazole drug delivery system [14]; and the production of an MIO-carboxymethyl cellulose composite by a co-precipitation method and the evaluation of its application as a potential drug delivery vehicle in the adsorption and desorption of a tetracycline antibiotic [20]. 

As MIO-NPs have several unique advantages in the targeted drug delivery system, such as efficient drug solubility and stability, accurate dosage, drug release controllability, and effective delivery of therapeutics to specific sites with low toxicity and side effects [16,21], the incorporation of MIO-NPs into the cellulose matrix would produce excellent non-toxic NCPs, potentially expanding the application and opportunities in biomedicine and nanotechnology. In addition, the antioxidant profiling of a proposed drug delivery system is required to check whether the synthesized nanomaterials are capable of reducing reactive oxygen species (ROS) generation or not [22]. To the best of our knowledge, the antioxidant properties and drug delivery applications of WTP/MIO-NCPs and SCB/MIO-NCPs obtained from WTP and SCB have not been investigated. The purpose of this study was to prepare MIO-incorporated WTP/MIO-NCPs and SCB/MIO-NCPs using the in situ co-precipitation method, as well as to evaluate the potentiality of composite materials as drug delivery vehicles in terms of swelling rate, drug-loading, and drug-releasing behavior in the metronidazole drug delivery system and to investigate their DPPH scavenging activity.

## 2. Materials and Methods

### 2.1. Materials

Analytical reagent-grade chemicals were purchased and used without further purification. Sodium hydroxide (NaOH), sodium hypochlorite (NaClO), sulphuric acid (H_2_SO_4_, assay ≥ 98%), acetic acid (CH_3_COOH, assay ≥ 99%), acetone (C_3_H_6_O, assay ≥ 99.5%), anhydrous ferric chloride (FeCl_3_), ferrous sulfate (FeSO_4_·7H_2_O), ammonium hydroxide (NH_4_OH), 2,2-diphenyl-1-picrylhydrazyl (DPPH), ascorbic acid (C_6_H_8_O_6_, assay ≥ 99.5%), methanol (CH_3_OH, assay ≥ 99.8%), disodium hydrogen phosphate (assay ≥ 99%), and sodium di-hydrogen phosphate (assay ≥ 99%) were purchased from Merck, Darmstadt, Germany, and deionized water was collected from RCI Labscan Ltd., Bangkok, Thailand. SCB and WTP were collected from a local juice store and brought to the laboratory for experimental studies.

### 2.2. Preparation of Cellulose-WTP Particles from WTP 

The extraction of cellulose-WTP particles from WTP was carried out based on the procedure outlined by Danial et al. [23] and Sheltami et al. [24], wherein the basic and acidic hydrolysis methods were followed. Briefly, the collected WTP was shredded into small pieces and boiled for 6 h, followed by the addition of distilled water periodically. It was then ground to obtain a slurry and filtered and rinsed several times with distilled water. Afterwards, the slurry was boiled again and treated with 5% sodium hydroxide (*w/v*), followed by the treatment with 2% sodium hypochlorite (*v/v*). Finally, the slurry was filtered, and the cellulose material was washed several times with deionized water until the pH became neutral. Thereafter, the pre-treated cellulose was constantly stirred for 1 h at 45 °C and 250 rpm on a hotplate with a magnetic stirrer (MS-H380-Pro DLAB, Rowland St. City of Industry, CA, USA) with the addition of 60% (*v/v*) H_2_SO_4_ acid solution, where the pretreated solution to acid ratio was 1:2 (*v/v*). The mixture was diluted with deionized water and then centrifuged at 2000 rpm (LC-8, Benchmark Scientific, Sayreville, NJ, USA) for 1 h to eliminate the spent acid, and this process was repeated three times. Subsequently, the suspension was dialyzed with deionized water until it attained a constant pH, and it was finally dried in an oven at 70 °C overnight. The overall process is summarized in Figure 1.

### 2.3. Preparation of Cellulose-SCB Particles from SCB 

The cellulose-SCB particles were extracted from the SCB following the delignification and bleaching, as previously discussed by Rashid et al. [8] and Evans et al. [25]. At first, the SCB was washed with deionized water, and then, it was sun-dried. Afterwards, it was crushed into small pieces using a commercial blender and sieved through double mesh sieves. The SCB powder was dried in an oven for 5 h at 105 °C. Afterwards, about 30 g of the bagasse powder was mixed with 300 mL of 6% HNO_3_ acid (*w/v*), and it was continuously stirred for 2 h at 80 °C and 250 rpm on a hotplate with a magnetic stirrer. The mixture was washed several times with deionized water until a neutral pH was achieved, and it was refluxed in the presence of 200 mL of 1% NaOH at 100 °C for 2 h on a magnetic hotplate with constant stirring at 250 rpm. Thereafter, the mixture was subsequently washed with deionized water and bleached with 100 mL of 0.735% sodium hypochlorite (*w/v*). The cellulosic extract was again subjected to acetic acid hydrolysis at 80 °C for 2 h. The mixture was further washed with deionized water until attaining a neutral pH, and finally, the extracted material was dried in an oven (ON-01E Jeio Tech, Seoul, Korea) at 80 °C. The pre-treated cellulose was hydrolyzed with 32% of the H_2_SO_4_ acid (*w/v*) under constant stirring for 24 h at 250 rpm and room temperature, where the cellulose to dilute acid ratio was 1:25 (*w/v*). This reaction was quenched by adding 10-fold more deionized water to the reaction mixture. The solution was centrifuged three times at 10,000 rpm for 15 min to eliminate the acid solution. The supernatant was discarded, and the cellulosic precipitate was redistributed in deionized water. Afterwards, it was subsequently dialyzed with deionized water, and the cellulose crystal was homogenized by sonicating the colloidal suspension for 1 h using a WD-9415A digital ultrasonic cleaner (Beijing, China). The generated crystals were further centrifuged at 6000 rpm for 30 min and kept for 24 h to settle the water, which was further replaced with acetone, followed by centrifugation at 6000 rpm for 30 min, and finally dried in an oven at 70 °C overnight. The overall preparation process is briefly depicted in Figure 2.

### 2.4. Preparation of WTP/MIO-NCPs and SCB/MIO-NCPs

The preparation of WTP/MIO-NCPs and SCB/MIO-NCPs was carried out based on the partially modified method of Rashid et al. [18], and the overall protocol is summarized in Figure 3. Briefly, 1.0 g of cellulose particles obtained from the WTP and SCB were separately transferred into a three-necked, round-bottomed flask that was dipped in an oil bath and dispersed in 50 mL of deionized water by magnetic stirring at 400 rpm for 30 min. The dispersion mixture was heated for 20 min at 60 °C, and then, 0.2 g of FeSO_4_·7H_2_O and FeCl_3_ (1:2 molar ratio) were added. The mixture was continuously stirred at 250 rpm for another 10–20 min, and 25% of the NH_4_OH solution was added. The reaction was continued in an inert atmosphere for more than two hours. At the end of the reaction, a black-colored composite material was precipitated, which was further magnetically separated, washed with deionized water, and then dried in an oven at 50 °C for 3 h.

### 2.5. Synthesis of MIO-NPs

MIO-NPs were synthesized based on the in situ co-precipitation of Fe^2+^ and Fe^3+^ ions in an alkaline medium, as described by Rashid et al. [8]. About a 1:2 molar ratio of FeSO_4_·7H_2_O (0.86 g) and FeCl_3_ (1.0 g) was taken into a 250 mL three-necked round-bottomed flask that was placed in an oil bath, and 50 mL deionized water was added. The calculated amount of 25% NH_4_OH (17.29 g) was added, and the mixture was stirred on a hotplate with a magnetic stirrer at 80 °C at 200 rpm in an inert atmosphere. Finally, the precipitated particles were magnetically separated, washed, and dried in an oven at 90 °C overnight.

### 2.6. Characterization

The synthesized cellulosic materials, MIO-NPs, WTP/MIO-NCPs, and SCB/MIO-NCPs, were characterized using various instruments, including a Fourier-transform infrared (FTIR) spectrophotometer (IRTracer-100, SHIMADZU, Asia Pacific Pte. Ltd., Singapore), an FESEM coupled with an energy dispersive X-ray (EDX) (JSM-6510, JEOL, Tokyo, Japan), and an XRD (Philips PANalytical X’PERT PRO, Tokyo, Japan). The magnetic property was inspected by employing a VSM (MicroSense, EV9, East Lowell, MA, USA). Moreover, the paramagnetic properties of the washed magnetic nanocomposite particles were studied using the Sherwood magnetic susceptibility balance (MK-I, Cambridge, UK). The magnetic susceptibility (*Ӽg*) was calculated according to Equation (1), where *C* denotes the calibration constant, *L* is the length of the sample (cm), *R*_0_ and *R* are the readings of the empty and sample tubes, and *M* means the weight of the sample in grams.
(1)$$Ӽg=\frac{C\times L\times \left(R-R_{0}\right)}{m\times 10^{9}}\;$$

### 2.7. Application of Cellulose, NPs, and NCPs

#### 2.7.1. Free Radical Scavenging Activity

The DPPH scavenging assay is the most widely used method for measuring anti-radical activity due to its rapidness, stability, and simplicity of analysis [26]. Herein, the antioxidant activity of the cellulose-WTP, cellulose-SCB, MIO-NPs, SCB/MIO-NCPs, and WTP/MIO-NCPs was measured against the DPPH according to the protocol outlined by Kundu et al. [27]. Initially, the DPPH was dissolved in a solution of methanol, and the concentration was adjusted to 0.007886% (*w*/*v*). About 10 µL of different concentrations (16, 32, 64, 128, and 256 μg/mL) of dispersed particles and ascorbic acid were taken, and 190 µL of DPPH in methanol was added to them, and ascorbic acid was used as a reference compound. The mixture was then allowed to remain at 25 °C for 30 min with vigorous shaking. Furthermore, the absorbance was measured using a UV-Vis spectrophotometer (UV-1900i, SHIMADZU, Asia Pacific Pte. Ltd.) at 517 nm. The lower value of the absorbance indicated higher free radical scavenging activity. However, the ability to scavenge the DPPH free radical was determined by Equation (2) [10], where *A*_0_ denotes the absorbance of the control reaction, and *A_1_* indicates the absorbance in the presence of either the sample or the reference material.
(2)$$Scavenged\;\left(\%\right)=\frac{\left(A_{0}-A_{1}\right)}{A_{0}}\times 100\;$$

#### 2.7.2. Swelling Profile Analysis

The method suggested by Azizi [14] was followed for measuring swelling, with partial modifications. Initially, raw tea was separated from the tea bags, and 0.1 g of SCB/MIO-NCPs, WTP/MIO-NCPs, cellulose-SCB, and cellulose-WTP were poured into them separately, and their weight was recorded. Thereafter, each tea bag was placed in a beaker containing 200 mL of phosphate buffer solution (pH = 7.4). After the time intervals (30–240 min), the bag was detached from the container and hung for 5 min to drain the excess buffer solution as well as to attain a constant weight. Finally, the wet weight of the bag was measured, and the equilibrium swelling ratio was calculated using Equation (3) where *W_i_* is the initial weight of the dried sample and *W_t_* is the swollen weight of the sample at time *t* [28].
(3)$$Swelling\;rate\;\left(\%\right)=\frac{\left(W_{t}-W_{i}\right)}{W_{i}}\times 100\;$$

#### 2.7.3. Drug Loading and Releasing Studies

The drug-loading and drug-releasing efficiency of cellulose-SCB, cellulose-WTP, WTP/MIO-NCPs, and SCB/MIO-NCPs were performed following the method outlined by Azizi [14], where metronidazole antibiotic was used as a model drug. For the drug-loading test, 0.2 g of each of the prepared nanostructures was added to 20 mL of phosphate buffer solution (pH 7.4) containing 150 mg/L metronidazole at room temperature. These were kept for 3 days to load the drugs into the cellulose and NCPs. The amount of loaded drug was estimated by measuring the remaining concentration of metronidazole in solution using UV–Vis spectroscopy at a maximum of 320 nm wavelengths after separating the particles from the supernatant by an external magnet or by withdrawing the filtered samples. Before determining the residual concentration, a standard calibration curve of various known concentrations of metronidazole was constructed, as described in the quality control section. The quantity of metronidazole drug loading was calculated based on Equation (4), where *M*_1_ represents the amount of metronidazole drug in the nanostructure, and *M*_0_ denotes the initial amount of the nanostructure.
(4)$$Drug\;loading\;\left(\frac{mg}{mg}\right)=\frac{M_{1}}{M_{0}}\;$$

The metronidazole drug-releasing rates of cellulose-SCB, cellulose-WTP, WTP/MIO-NCPs, and SCB/MIO-NCPs were determined in a phosphate buffer solution of pH 7.4 at 37 °C without applying any mixing conditions. For this assessment, 0.01 g of nanostructure loaded with the metronidazole drug was immersed in 50 mL of buffer solution. At certain times, 2 mL of each solution was ejected, and after the separation of the particles, the drug concentration was quantified. The drug release (%) was calculated according to the following Equation (5), where *C_t_* represents the released amount of metronidazole at time *t*, and *C_i_* refers to the initial loaded metronidazole amount.
(5)$$Drug\;release\;\left(\%\right)=\frac{C_{t}}{C_{i}}\times 100\;$$

### 2.8. Quality Control

The analytical method was validated according to the prescribed guidelines of ICH in terms of linearity, sensitivity, reproducibility, and stability measurements [29]. 

#### 2.8.1. Linearity

Linearity was established by the least squares linear regression analysis of the calibration curve. Initially, a standard stock solution (100 μg/mL) was prepared by adding 10 mg of metronidazole drug to 100 mL of buffer solution (pH 7.4). Then, a series of standard solutions (1–10 μg/mL) were prepared by diluting the stock solution, and the absorbance for each solution was measured at 320 nm. Afterwards, a standard calibration curve was drawn by plotting a graph of absorbance vs. concentration (Figure 4). The values of the correlation coefficient (R^2^) and the standard regression equation were found to be 0.9987 and y = 0.0624x – 0.0035, respectively.

#### 2.8.2. Sensitivity

The limit of detection (*LOD*) and the limit of quantification (*LOQ*) indicate the sensitivity of an analytical method. The *LOD* is the lowest concentration of an analyte detected by the method, whereas the *LOQ* is the minimum quantifiable concentration [30]. In this study, the *LOD* and the *LOQ* were calculated from the standard calibration curve using Equations (6) and (7). The values of the *LOD* and the *LOQ* were estimated as 0.380 μg/mL and 1.151 μg/mL, based on the residual standard deviation (*SD*) of the regression line, and 0.259 μg/mL and 0.786 μg/mL from the *SD* of the Y-intercept of the regression line, respectively (Table 1).
(6)$$LOD=\frac{3.3\times SD}{Slope}\;$$
(7)$$LOQ=\frac{10\times SD}{Slope}\;$$

#### 2.8.3. Reproducibility

The reproducibility of the method was examined by intra-day and inter-day variation studies. Intra-day precision was demonstrated by analyzing the absorbance of a 5 µg/mL drug sample three times per day, while inter-day precision was verified by investigating the same samples three times for three consecutive days. The acceptable limit of the relative standard deviation (%*RSD*) for intra-day and inter-day variation should be within 1% and 2%, respectively [31]. In this study, the %*RSD* for intra-day and inter-day reproducibility was calculated by Equation (8), where $\bar{x}$ indicates the mean absorbance of the sample, and the results were found to be below 1%; hence, the method is reproducible (Table 2).
(8)$$\%RSD=\frac{SD}{\left|\bar{x}\right|}\times 100\;$$

### 2.9. Statistical Analyses

For each sample, the measurements were made in triplicate, and the results were presented as means ± SD. The data were statistically analyzed using GraphPad Prism software version 8.0.1. A one-way analysis of variance (ANOVA) technique with a *t*-test (*p*-value *≤* 0.05) was employed to examine the significant differences in the performances of the synthesized nanomaterials.

## 3. Results and Discussion

### 3.1. Characterization

#### 3.1.1. FTIR Analysis

FTIR is one of the most commonly used instrumental techniques for the identification of the different functional groups present in the nanocomposites, and the FTIR spectra of the cellulose, NPs, and NCPs are displayed in Figure 5. The FTIR spectra of cellulose obtained from the WTP and SCB exhibited characteristic bands at 3406 cm^−1^ (2998–3600 cm^−1^), indicating the tensile vibrations of O–H groups, while the absorption peaks near 1430 cm^−1^, 1373 cm^−1,^ and 1064 cm^−1^ revealed the presence of the tensile and flexural vibrations of the C–H group and the tensile vibration of the C–O group, respectively [14]. The signal around 1642 cm^−1^ appeared due to the cumulative vibration of the O–H bending of the absorbed water [10]. The bands at 1064 cm^−1^ and 897 cm^−1^ were attributed to the skeletal vibration of the –C–O–C– pyranose ring and the symmetric stretching of glycosidic linkages between the glucose units [10,13,25]. The absorption peaks that appeared in the pure cellulose spectra were also noticed in the FTIR spectra of the WTP/MIO-NCPS and SCB/MIO-NCPs. It is noteworthy that all the spectra were mostly similar except for some additional signals or shifts in the original peak position. The signal near 2906 cm^−1^ in the spectrum of cellulose, WTP/MIO-NCPS, and SCB/MIO-NCPs originated from the C–H stretching and rocking vibrations of the methylene groups (–CH_2_) [13]. In the FTIR spectra of the MIO-NCPs, WTP/MIO-NCPS, and SCB/MIO-NCPs, the presence of absorption bands near 583 cm^−1^, 577 cm^−1,^ and 572 cm^−1^, respectively, indicated the flexural vibrations of Fe–O bonds in the partially disordered MIO particles [13,14]. This finding revealed the impregnation of MIO-NPs in the cellulose matrix during the formation of NCPs. Similar types of absorption bands for cellulose were previously noticed by Lu et al. [1], Rashid et al. [8], Rabbi et al. [10], Azizi [14], and Khodaei et al. [32] while investigating the structure of metal-incorporated cellulose NCPs.

#### 3.1.2. FESEM and EDX Analysis

The SEM images of the cellulose, MIO-NPs, WTP/MIO-NCPs, and SCB/MIO-NCPs are shown in Figure 6. The aggregation of cellulose was observed in the microscopic image (Figure 6a), which might be due to the polar nature of cellulose particles or the gradual evaporation of the dispersion medium, which brought the single cellulose crystallites closer and facilitated the accumulation through hydrogen bonds [13,33,34,35]. The morphology of the synthesized MIO-NPs (Figure 6b) was found to be irregularly spherical and agglomerated in nature, as previously mentioned in similar articles [36,37]. As the MIO-NPs showed magnetic properties, they could combine to produce aggregated particles. Moreover, this phenomenon might occur due to the steric effect ascribed to the interaction between the magnetic NPs [36,37]. In this study, the WTP/MIO-NCPs and SCB/MIO-NCPs were prepared by the in situ co-precipitation of cellulose particles and MIO-NPs in a basic medium, where the MIO-NPs were trapped in the cellulose matrix. Like the MIO-NPs, both the SCB/MIO-NCPs and the WTP/MIO-NCPs showed positive responses under the influence of an external magnetic field, which confirmed the magnetic property of the NCPs and resulted in agglomerated spherical-like porous particles (Figure 6c,d). However, the cellulose particles were enriched with hydroxyl groups, and these hydroxyl groups were expected to take part in the formation of cellulose/MIO hydrogen bonds [8].

The formation of MIO-NPs, WTP/MIO-NCPs, and SCB/MIO-NCPs was investigated by EDX analysis, and the relative elemental compositions (C, O, and Fe) are presented in Figure 7. In cellulose, the presence of carbon (C) and oxygen (O) was confirmed by the EDX analysis (Figure 7a). The sharp peaks at 6.4 keV and 0.5 keV in the EDX spectrum indicated the existence of elemental iron (Fe) and oxygen (O), respectively, for the MIO-NPs (Figure 7b) [36]. Similarly, the presence of intense peaks for elemental C, O, and Fe in Figure 7c,d confirmed the insertion of MIO-NPs into the cellulose matrix and the formation of SCB/MIO-NCPs and WTP/MIO-NCPs. 

#### 3.1.3. XRD Analysis

XRD analysis was employed to analyze the crystalline structure of the cellulose-WTP, cellulose-SCB, MIO-NPs, WTP/MIO-NCPs, and SCB/MIO-NCPs. In Figure 8, the cellulose particles exhibit two characteristic diffraction peaks at 16.2° and 22.5° for the (110) and (200) planes of the crystalline cellulose structure [1]. More specifically, the highest diffraction peak at 22.5° corresponds to the crystalline structure of cellulose I [23], whereas the low diffraction peak at 16.2° (near 18°) refers to the amorphous background [24]. The figure also showed the XRD pattern of the MIO-NPs, which had five distinct signals at 30.24°, 35.49°, 43.32°, 57.12°, and 62.79° for the corresponding (220), (311), (400), (511), and (440) planes, respectively. These diffraction signals were consistent with the JCPDS card no. 19–0629, which revealed that the synthesized MIO-NPs were crystalline [36]. Moreover, the absence of impurity phase peaks in the XRD spectrum indicated the single-phase structural nature and purity of the NPs, which had previously been reported in other similar studies [2,14,38]. The XRD peaks of WTP/MIO-NCPs and SCB/WTP-NCPs exhibited similar diffraction patterns for the cellulose and MIO-NPs, indicating that the MIO-NPs were anchored in the cellulose matrix and retained their phase properties [1]. It is notable that the diffraction signals in both the WTP/MIO-NCPs and the SCB/WTP-NCPs due to crystalline cellulose were not prominent and almost disappeared because of the existence of the hydrogen bonds and electrostatic interaction between the MIO-NPs and the cellulose particles that disrupt the crystal structure of cellulose. In addition, the poor diffraction signals may be due to the presence of MIO-NPs with a lower content [8]. Similar observations had also previously been reported by other researchers while preparing MIO-NPs-incorporated cellulose NCPs [1,38]. The average crystal size of the NPs and NCPs was calculated based on the full-width half maximum of the diffraction peaks by using the Debye–Scherrer formula [36,39]. The average crystallite size of the MIO-NPs, SCB/MIO-NCPs, and WTP/MIO-NCPs was obtained as 12.38 nm, 10.85 nm, and 11.47 nm, respectively, which was in line with the mean size of the nanocomposite (17 nm) and the MIO-NPs (28 nm), reported in the study of Azizi [14]. Equivalently, 15.28 nm, 7.21 nm, and 4.16 nm were also well-documented for MIO-NPs, mechanically activated pretreated cellulose, and FeCl_3_-incorporated mechanically activated pretreated cellulose, respectively [1]. 

#### 3.1.4. VSM Analysis

Magnetic property is an important feature of a magnetic material. In this study, the magnetic characteristics of magnetically separated nanomaterials were measured in a field of 20,000 to −20,000 at room temperature by VSM analysis. The hysteresis loops of the MIO-NPs and cellulose/MIO-NCPs are shown in Figure 9. The saturation magnetization (M_s_) is the maximum magnetic response of a material in the presence of an external magnetic field [10], which can be useful in the measurement of magnetic properties. In this experiment, the value of M_s_ for the MIO-NPs (32.43 emu/g) was comparatively higher than that of the cellulose/MIO-NCPs (31.83 emu/g), which was due to the strong interaction between the cellulose particles and MIO-NPs and the use of non-cellulosic raw material during the preparation of the NCPs. The findings of the present work comply with the behavior reported by Rabbi et al. [10], Azizi [14], and Zhu et al. [40] while studying the magnetic property of this nanomaterial. However, the decreasing value of M_s_ also indicated that the cellulose/MIO-NCPs possessed weaker magnetic properties than the MIO-NPs. For both the NPs and the NCPs, almost zero coercivity and the S-shaped reversible magnetization curve were noticed, confirming the existence of the paramagnetic nature of the synthesized nanomaterials [2,41].

### 3.2. Applications of Cellulose, NPs, and NCPs

#### 3.2.1. DPPH Scavenging Assay Analysis

Free radicals are extremely reactive atoms, molecules, or ions that have one or more unpaired electrons, produced during oxidation reactions. These species might be hazardous to the body and cause several health implications, such as heart diseases, cancer, stroke, and aging-related ailments. The highly reactive oxygen species are generated during mitochondrial oxidative metabolism [42]. An excess of ROS leads to oxidative stress that results in cellular damage or abnormal cell growth. Anti-oxidizing agents help in scavenging these free radicals from the bodily cells and in preventing the damage caused by their chain reactions [43]. DPPH is a well-known toxic free radical that can pose detrimental effects on human health. The stability of this reactive species arises by virtue of the delocalization of the free electron over the molecule as a whole [44]. In the presence of an antioxidant molecule or metallic particles, the uncharged DPPH free radical can be altered into the non-radical form by accepting a hydrogen atom or an electron; thus, it has long been used to test the free radical scavenging capacity of antioxidants [8,44,45]. In this current study, the antioxidant activity of cellulose-WTP, cellulose-SCB, MIO-NPs, WTP/MIO-NCPs, and SCB/MIO-NCPs at different concentrations was determined by DPPH radical scavenging assay, and the obtained result is illustrated in Figure 10. Furthermore, the percentage (%) quenched of a standard reference compound (ascorbic acid) was compared. The higher the percent scavenged, the better the antioxidant ability of the nanostructures. 

The findings of the DPPH assay indicated that the cellulose-WTP and cellulose-SCB did not show any scavenging activity, while the WTP/MIO-NCPs, SCB/MIO-NCPs, and MIO-NPs exhibited almost negligible antioxidant activity in comparison to ascorbic acid. In addition, no significant difference (*p*-value ≤ 0.05) was noticed between the MIO-NPs, the WTP/MIO-NCPs, and the SCB/MIO-NPs. In a study, Rabbi et al. [8] showed that neither cellulose particles obtained from jute fiber nor MIO-incorporated cellulose particles exhibited antioxidant activity due to their identical behavior, and the antioxidant capacity of MIO-NPs could hardly be observed due to their negligible % scavenged value, which complies with our findings. 

However, MIO-NPs can quench the DPPH free radicals in a dose-dependent manner, which is possibly due to the neutralization of the free radical by the transfer of an electron [46]. It should be noted that NPs such as Ag, Au, CuO, ZnO, Pt, Fe_3_O_4_, Co_3_O_4_, etc., particularly those made using biological or green methods, have a significant antioxidant ability [22,44,45,46,47,48,49]. The presence of several phytochemicals, such as flavonoids, terpenoids, alkaloids, polyphenols, and poly-carboxylic acids, on the surface of MIO-NPs may be responsible for their boosted antioxidant efficiency [45]. In this current study, MIO-NPs, WTP/MIO-NCPs, and SCB/MIO-NCPs were prepared based on the co-precipitation method, but due to the absence of phytoconstituents on the surface of the synthesized nanostructures, negligible antioxidant activity was observed. 

#### 3.2.2. Swelling Behavior Analysis

The swelling behavior plays an essential role in the mass transfer of water into and out of the matrix and thus affects the release kinetics of the incorporated materials [50]. In this study, the swelling behavior of the cellulose-WTP, cellulose-SCB, SCB/MIO-NCPs, and WTP/MIO-NCPs was analyzed for 240 min and is illustrated in Figure 11.

The swelling rates of the SCB/MIO-NCPs and WTP/MIO-NCPs were rapidly increased up to 90 min and then slowed down until attaining the highest swelling rates of 155.0% and 159.5%, respectively. The swelling capacity of both NCPs was much higher than that of the cellulose-SCB (58.3%) and cellulose-WTP (61.6%), which might be due to the small size and large surface area of the MIO-NPs, resulting in the formation of an extensive specific surface area in the nanocomposite structure [14]. Moreover, the presence of MIO-NPs can enhance the network texture by increasing the number of pores, which can provide more space for the storage of solvent. Such observations have also been previously documented in similar studies, where Azizi [14] discussed the preparation of MIO-NPs and cellulose/MIO-NCPs via the green route, and Yadollahi et al. illustrated the synthesis method and characterization process of chitosan/ZnO-NCPs [51] and chitosan/Ag-NCPs [52].

#### 3.2.3. Drug Incorporation and Release Efficiency Analysis

In recent years, biopolymers have been widely used as raw materials for proposing suitable drug-delivery systems due to their excellent features, such as biodegradability, biocompatibility, non-toxicity, eco-friendliness, etc. [51,53]. In addition, there has been a great interest in generating organic–inorganic nanocomposites because of their superior biomedical applications. The introduction of NPs not only reduces the burst release effect but also increases the stability of the drug and provides a slower and more continuous release mode for drugs [51,52]. In this study, the drug-loading and drug-releasing efficiency of cellulose (WTP and SCB), MIO-NPs, WTPs/MIO-NCPs, and SCB/MIO-NCPs were evaluated in a metronidazole antibiotic drug delivery system, and the obtained results are presented in Figure 12. The order of the metronidazole drug loading after 3 days was: cellulose-SCB < cellulose-WTP < MIO-NPs < SCB/MIO-NCPs < WTP/MIO-NCPs, whereas the sequence of the drug-releasing rate after 240 min was: WTP/MIO-NCPs (14.52%) < SCB/MIO-NCPs (19.45%) < MIO-NPs (32.28%) < cellulose-WTP (45.88%) < cellulose-SCB (47.36%). 

The highest loadings of metronidazole occurred in the cellulose/MIO-NCPs, and the impregnation of the MIO-NPs in the cellulose matrix significantly (*p*-value ≤ 0.05) increased the drug-loading capacity of the WTP/MIO-NCPs and SCB/MIO-NCPs in comparison to the MIO-NPs and cellulose (WTP and SCB). This is probably due to the presence of small-size MIO-NPs in the cellulose matrix, which created a lot of free space and a porous structure. In addition, the incorporation of MIO-NPs could generate capillary forces that facilitate the penetration and adsorption of the drug molecules into the cellulose/MIO-NCPs [14]. It is noteworthy that the drug-loading capability of the MIO-NPs was slightly higher than that of the cellulose obtained from the WTP and SCB, although there was no significant difference (*p*-value ≤ 0.05) among them. However, a greater loading capability of MIO-NPs over cellulose particles may be achieved due to their larger surface area, small size, and shape. In a study, Vangijzegem et al. [17] reported that MIO-NPs can be used in targeted drug delivery systems either as individual NPs or NCPs by controlling the size and shape carefully.

When the drug-releasing behavior was investigated, it was noticed that the release rate in the WTP/MIO-NCPs and SCB/MIO-NCPs was significantly (*p*-value ≤ 0.05) slower than that of pure cellulose (SCB and WTP). This phenomenon can be explained by the fact that metronidazole was tightly held within the pores of the NCPs by the strong electrostatic interactions and hydrogen bonding between the drugs and particle or pore surfaces at neutral pH [54]. Moreover, the overall drug release is affected by the rate of solvent uptake and the diffusion rate of the drug. As there is an inverse relationship between the drug release rate and the matrix swelling rate [55], a slower drug release was observed in the NCPs. However, the obtained result of the drug-loading and drug-releasing rate has been compared with similar studies, which comply with our present findings (Table 3). According to Yadollahi et al. [51,52] and Azizi [14], the incorporation of Ag-NPs, ZnO-NPs, and MIO-NPs in pure chitosan beads (hydrogel) and cellulose, respectively, was responsible for prolonged drug-releasing behavior. 

## 4. Conclusions

In summary, the current study reported the synthesis processes of WTP/MIO-NCPs and SCB/MIO-NCPs from WTP and naturally abundant SCB. In addition, the structural, morphological, and magnetic characteristics of the prepared nanomaterials were inspected using several advanced techniques, such as FTIR, XRD, FESEM, EDS, and VSM, respectively. The FESEM, XRD, and VSM analyses revealed that the synthesized materials were agglomerated, irregularly spherical, and nanocrystalline in size and that they retained magnetic properties. Although the WTP/MIO-NCPs, SCB/MIO-NCPs, and MIO-NPs showed negligible antioxidant activity compared to the ascorbic acid, they exhibited a better swelling efficiency, a higher drug-loading capacity, and a slower drug-releasing rate than that of the cellulose obtained from SCB and WTP. The overall findings suggested that MIO-incorporated cellulose particles could be a promising material for the metronidazole drug-delivery system. 

However, this study has several limitations, which should be taken into consideration in future studies. Some of them are as follows: (a) the effect of pH, temperature, drug-releasing time (3 days), burst release, and the variation of the doses of cellulose, NPs, and NCPs on the investigated system was not assessed; (b) in vivo experiments regarding the cytotoxic activity of synthesized particles were not examined; (c) metronidazole is often used in combination with other drugs, but drug loading and drug release in the dual and multi-drug delivery systems were not studied. 

## Figures and Tables

**Figure 1 pharmaceutics-15-00732-f001:**
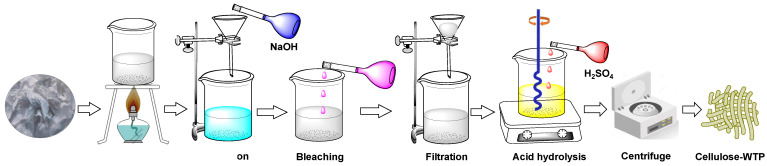
Preparation of cellulose-WTP particles from WTP.

**Figure 2 pharmaceutics-15-00732-f002:**
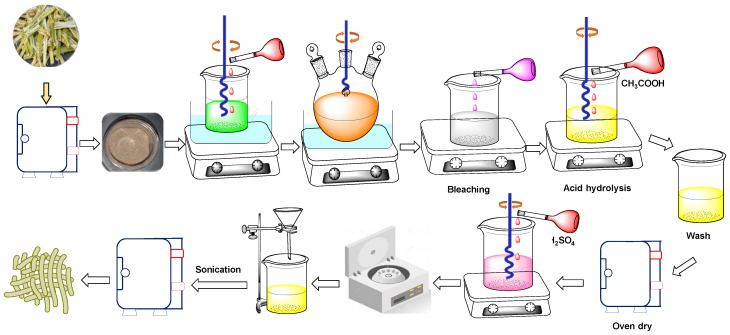
Preparation of cellulose-SCB particles from SCB.

**Figure 3 pharmaceutics-15-00732-f003:**
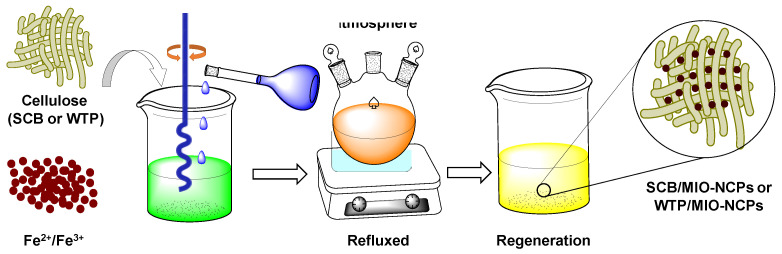
Overall preparation protocols of WTP/MIO-NCPs and SCB/MIO-NCPs.

**Figure 4 pharmaceutics-15-00732-f004:**
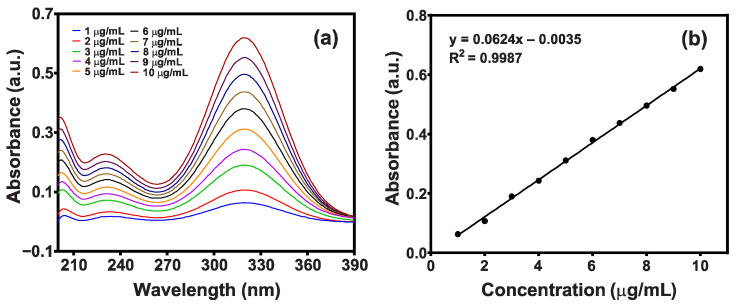
(**a**) Scanning curve and (**b**) calibration curve of standard metronidazole drug in a buffer solution of pH 7.4.

**Figure 5 pharmaceutics-15-00732-f005:**
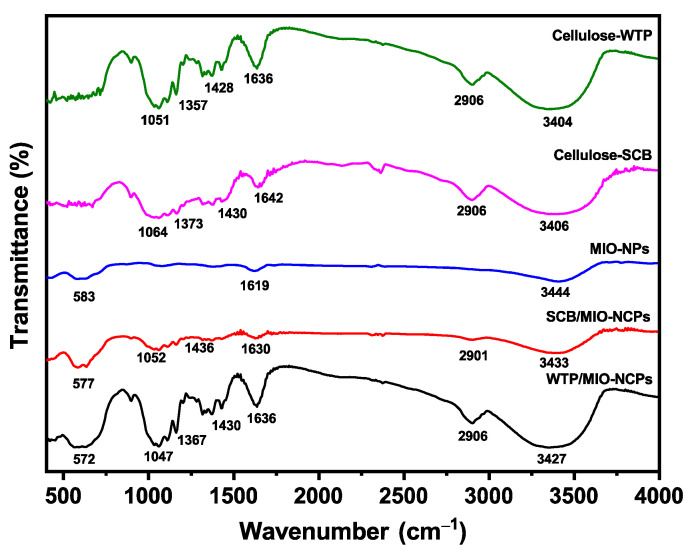
FTIR spectra of cellulose-SCB, cellulose-WTP, MIO-NPs, WTP/MIO-NCPs, and SCB/MIO-NCPs.

**Figure 6 pharmaceutics-15-00732-f006:**
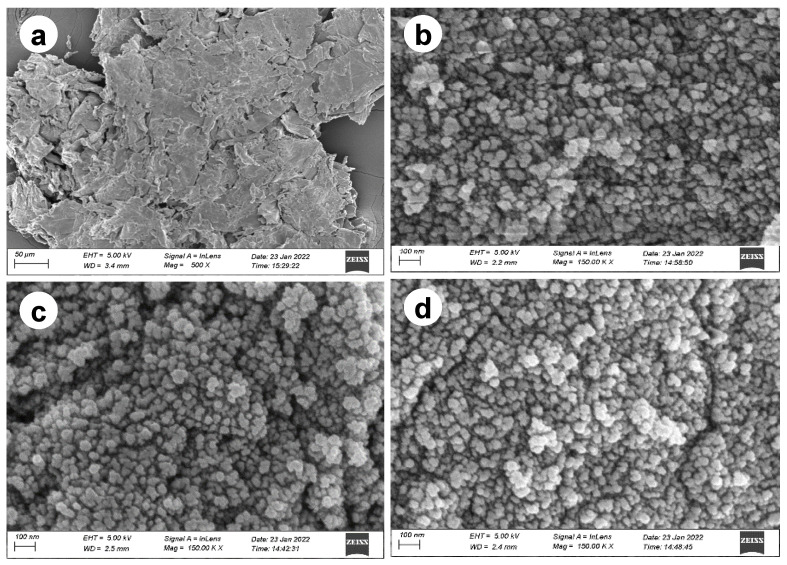
SEM image of (**a**) cellulose, (**b**) MIO-NPs, (**c**) SCB/MIO-NCPs, and (**d**) WTP/NCPs.

**Figure 7 pharmaceutics-15-00732-f007:**
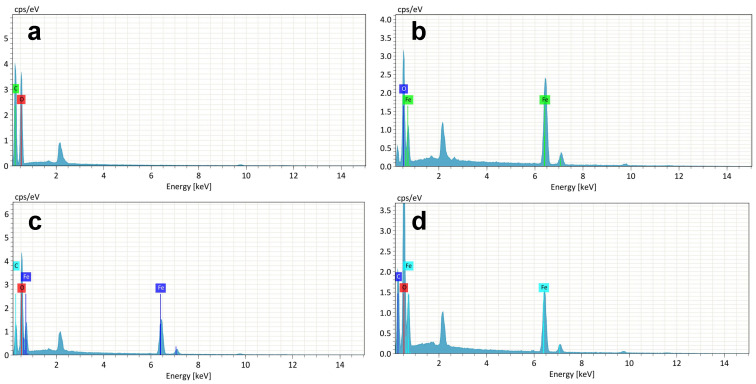
EDS spectrum of (**a**) cellulose, (**b**) MIO-NPs, (**c**) SCB/MIO-NCPs, and (**d**) WTP/MIO-NCPs.

**Figure 8 pharmaceutics-15-00732-f008:**
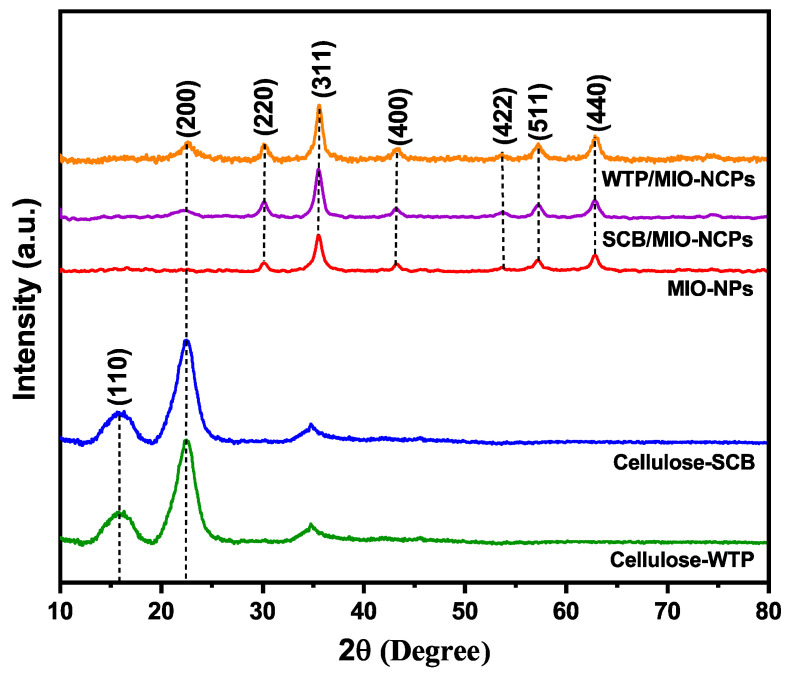
XRD patterns of cellulose-WTP, cellulose-SCB, SCB/MIO-NCPs, WTP/MIO-NCPs, and MIO-NPs.

**Figure 9 pharmaceutics-15-00732-f009:**
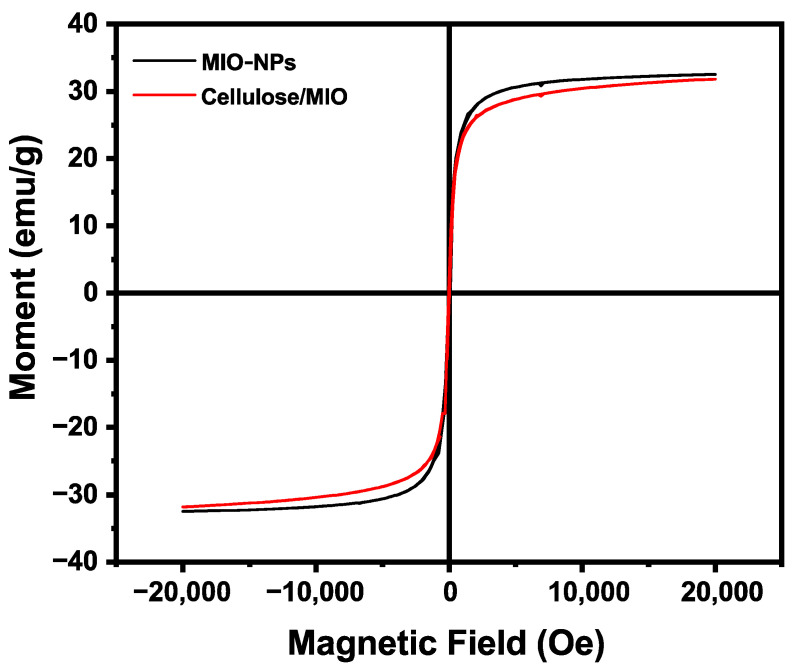
Magnetic properties of MIO-NPs and Cellulose/MIO-NCPs.

**Figure 10 pharmaceutics-15-00732-f010:**
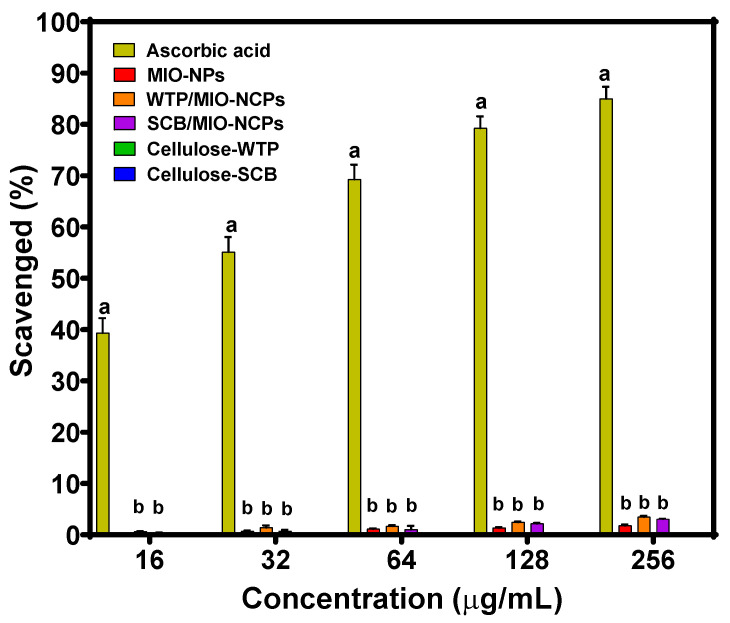
DPPH scavenging assay of ascorbic acid (control), MIO-NPs, WTP/MIO-NCPs, SCB/MIO-NCPs, cellulose-WTP, and cellulose-SCB. Similar letters above bars indicate no significant differences at *p*-value ≤ 0.05.

**Figure 11 pharmaceutics-15-00732-f011:**
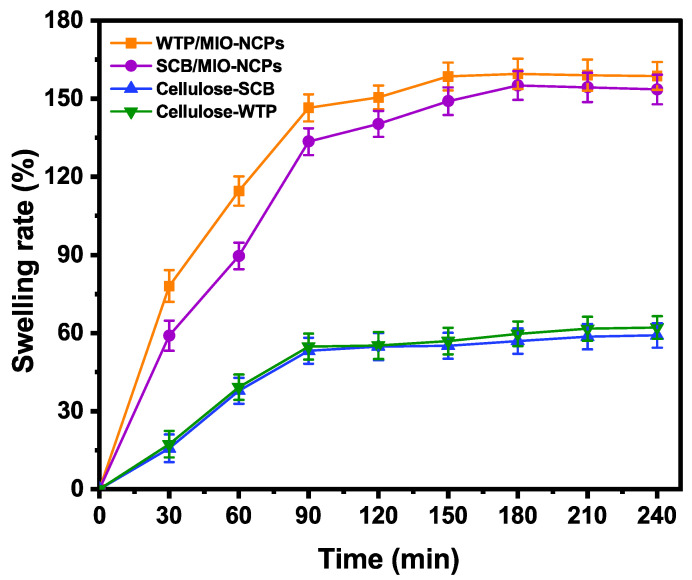
The swelling rates in cellulose-SCB, cellulose-WTP, WTP/MIO-NCPs, and SCB/MIO-NCPs.

**Figure 12 pharmaceutics-15-00732-f012:**
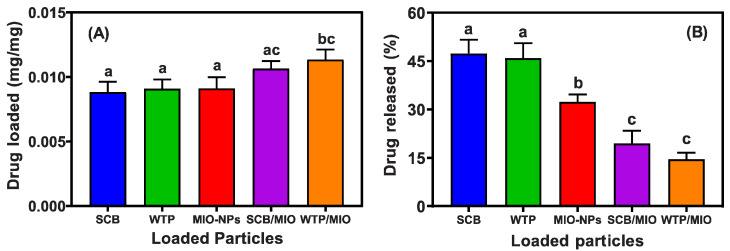
(**A**) Amount of drug-loading and (**B**) drug-releasing efficiency of cellulose (SCB and WTP), MIO-NPs, SCB/MIO-NCPs, and WTP/MIO-NCPs. Similar letters above bars indicate no significant differences at *p*-value ≤ 0.05.

**Table 1 pharmaceutics-15-00732-t001:** *LOD* and *LOD* for sensitivity analysis.

Parameter	From the Residual *SD* of the Regression Line (µg/mL)	From the *SD* of the Y-Intercept of the Regression Line (µg/mL)
*LOD*	0.380	0.259
*LOQ*	1.151	0.786

**Table 2 pharmaceutics-15-00732-t002:** Intra-day and inter-day reproducibility analysis.

Sl. No.	Conc. (µg/mL)	Intra-Day Precision			Inter-Day Precision		
		10 am	1 pm	4 pm	Day-1	Day-2	Day-3
1	5	0.304	0.307	0.308	0.304	0.310	0.313
2	5	0.304	0.308	0.307	0.304	0.311	0.312
3	5	0.304	0.306	0.309	0.304	0.312	0.313
4	5	0.306	0.308	0.307	0.306	0.312	0.314
5	5	0.305	0.308	0.308	0.305	0.311	0.314
6	5	0.306	0.307	0.309	0.306	0.310	0.315
% *RSD*		0.323	0.266	0.290	0.323	0.288	0.335

**Table 3 pharmaceutics-15-00732-t003:** Comparison of drug loading (mg/mg) and drug release (%) with similar studies.

Drug Delivery Systems	Drug Loaded (mg/mg)	Releasing Time (min)	Drug Released (%)	References
Chitosan/Ag-0	0.00941	1440	29.85	[52]
Chitosan/Ag-1	0.00814		23.15	
Chitosan/Ag-2	0.00750		16.23	
Chitosan/Ag-3	0.00728		8.96	
Chitosan/ZnO-0	0.00941	1440	29.85	[51]
Chitosan/ZnO-1	0.00996		11.49	
Chitosan/ZnO-2	0.00995		7.11	
Chitosan/ZnO-3	0.00998		5.7	
Pure cellulose	0.00901	180	42	[14]
MIO-NPs	0.00996		36	
MIO-NPs/cellulose	0.01210		15	
Cellulose (SCB)	0.00880 ± 0.00082	240	47.36 ± 4.26	Present study
Cellulose (WTP)	0.00907 ± 0.00074		45.88 ± 4.67	
MIO-NPs	0.00910 ± 0.00089		32.28 ± 2.34	
SCB/MIO-NCPs	0.01063 ± 0.00061		19.45 ± 3.92	
WTP/MIO-NCPs	0.01133 ± 0.00083		14.52 ± 2.07	

## Data Availability

Not applicable.

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
