# Peer review of "Synthesis of Magnetic Iron Oxide-Incorporated Cellulose Composite Particles: An Investigation on Antioxidant Properties and Drug Delivery Applications"

_pharmaceutics, 2023, doi:10.3390/pharmaceutics15030732_

Round 1

Reviewer 1 Report

In the presented study, the magnetic iron oxide incorporated cellulose composite particles were synthesized and the antioxidant and drug delivery properties were investigated. This study showed that the incorporation of MIO-NPs in the cellulose matrix increased the swelling capacity, drug-loading capacity, and drug-releasing time. However, the manuscript must be revised for publishing on Pharmaceutics considering the following remarks:

1. In line 24, what is NPs?

2. In lines 59-82, it should be rewrote and the author should mainly introduce the development and problems of magnetic iron oxide incorporated nanocomposite particles in the field of drug delivery.

3. In Figures 3,4,5 and 7, where was cellulose obtained from? And in the manuscript the author should show the characterization results of cellulose from SCBs and WTPs.

4. From SEM result, the particle size or shape of MIO-NPs, SCB/MIO-NCPs and WTP/NCPs had no obvious difference, the presentation from lines 280 to 296 should be discussed again. 

5. In lines 318-321, what substance do the typical diffraction belong to? And the substance should be marked in figure 7. Do there have iron hydroxide under the experiment temperature?

6. From figure 7, why was the typical diffraction peak of (311) more stronger after the iron nano-particles was coated by cellulose?

7. In figure 9, why was DPPH scavenging of WTP/MIO much higher than that of SCB/MIO? Why was the initial concentration of 16 ug/mL selected?

8. From Table 2, why was the drug loading of MIO-NPs much higher than that of cellulose?

9. In lines 447-462, the presentation was unreasonable, it must be discussed again.

10. In the manuscript, what were the relationship among the swelling capacity, the antioxidant property and drug delivery property? The effect of the swelling capacity and the antioxidant property to the drug delivery property should be discussed in the paper.

11. The English and the tense of the manuscript should be improved.

Author Response

Reply to Reviewer #1

Comment-1. 1. In line 24, what is NPs?

Reply-1: Accepted. NPs denote nanoparticles, and we have corrected this (Current line 16).

Comment-2. In lines 59-82, it should be rewrote and the author should mainly introduce the development and problems of magnetic iron oxide incorporated nanocomposite particles in the field of drug delivery.

Reply-2: Accepted and discussed those lines based on your valuable suggestion (Current line 73-109).

Comment-3. In Figures 3,4,5 and 7, where was cellulose obtained from? And in the manuscript the author should show the characterization results of cellulose from SCBs and WTPs.

Reply-3: Accepted and corrected by providing the relevant information in the figure (3, 5, and 8).

Comment-4. From SEM result, the particle size or shape of MIO-NPs, SCB/MIO-NCPs and WTP/NCPs had no obvious difference, the presentation from lines 280 to 296 should be discussed again.

Reply-4: Accepted and resolved the problems carefully (343-361).

Comment-5. In lines 318-321, what substance do the typical diffraction belong to? And the substance should be marked in figure 7. Do there have iron hydroxide under the experiment temperature?

Reply-5: Accepted. This time we have provided the diffraction pattern of SCB, WTP, MIO-NPS, SCB/MIO-NCPs, and WTP/MIO-NCPS. MIO-NPs indicated magnetic iron oxide, which exhibits characteristic diffraction peaks as shown in this figure (figure 8)( Current line 410-411).

Comment-6. From figure 7, why was the typical diffraction peak of (311) stronger after the iron nano-particles was coated by cellulose?

Reply-6: Thank you so much for your observation. Generally, MIO-NPs show a prominent peak at 311 planes. In crystalline cellulose, there is also a diffraction peak present near the 311 planes and after the incorporation of MIO-NPs in the cellulose matrix by the co-precipitation method, the peak at the 311 planes might be overlapped, as a result, the peak intensity might be increased. Another reason would be due to the variation of preparation protocols that can be explained as follows:

  1. Initially, cellulose particles from SCBs and WTPs were separately prepared by following different methods
  2. Then SCB/MIO-NCPs and WTP/MIO-NCPs were produced by adding the equimolar ratio of Fe2+/Fe3+ salts into SCB and WTP-cellulose particles separately, which is known as the co-precipitation method.
  3. MIO-NPs were prepared by mixing an equimolar ratio of Fe2+/Fe3+ salts in presence of ammonium hydroxide for confirming and comparing the characteristics MIO-NPs with SCB/MIO-NCPs and WTP/MIO-NCPs.

Comment-7. In figure 9, why was DPPH scavenging of WTP/MIO much higher than that of SCB/MIO? Why was the initial concentration of 16 µg/mL selected?

Reply-7: Accepted. In this study, cellulose was obtained and extracted from two different sources e.g. SCB and WTP by following two different methods, therefore a little variation may occur in their results. Besides, ANOVA analysis also showed that the incorporation of MIO-NPs in the cellulose matrix does not show any significant difference (at p-value <0.05 level) in the activity of WTP/MIO and SCB/MIO (Current line 456-460).

In our laboratory, we study the antioxidant activity based on 2, 4, 8, 16, 32, 64, 128, and 256 µg/mL where we consider 2 µg/mL as the lowest concentration. In this study, we did not find any antioxidant response for 2, 4, and 8 µg/mL, therefore, we presented data from 16 µg/mL. Although there was also a negligible response in the rest of the concentrations.

Comment-8. From Table 2, why was the drug loading of MIO-NPs much higher than that of cellulose?

Reply-8: Accepted. We have performed an analysis of variance (ANOVA) at <0.05 level to check the significant difference among the synthesized materials and found that there is no significant difference in the drug loading performance between MIO-NPs and Cellulose, although the drug loading value of MIO-NPs is slightly higher than that of cellulose. This might be due to the small size and larger surface area of MIO-NPs that cause greater adsorption (Current line 515-528).

Comment-9. In lines 447-462, the presentation was unreasonable, it must be discussed again.

Reply-9: Accepted. We have discussed this paragraph again based on your valuable suggestion (Current line 529-539).

Comment-10. In the manuscript, what were the relationship among the swelling capacity, the antioxidant property and drug delivery property? The effect of the swelling capacity and the antioxidant property to the drug delivery property should be discussed in the paper.

Reply-10: Accepted and discussed in the introduction (Current line 100-102) and section 3.3.3 (Current line 529-534).

Comment-11. The English and the tense of the manuscript should be improved.

Reply-11: Accepted. We carefully checked and improved the English and tense throughout the manuscript. Hope this time the manuscript will be appreciable.

Reviewer 2 Report

The authors investigate the preparation of iron oxide (MIO) loaded cellulose (WTP) and sugarcane bagasse (SCB) for metronidazole drug delivery systems. Metronidazole is an antibiotic and antiprotozoal medication that is commercially available. The characterization of the drug delivery system include Fourier Transform Infrared (FTIR) Spectroscopy, Field Emission Scanning Electron Microscopy (FESEM), Energy Dispersive Spectroscopy (EDS), Powder X-ray Diffraction (XRD), Vibrating Sample Magnetometry (VSM) and etc. After the characterization of the nanoparticles, there are not so many results for a paper in a pharmaceutical journal.

I have the following suggestions for a major review:

1.       The abstract should emphasize what is the novelty of the paper.

2.       The results shown on Fig.9 needs to be appended with pure WTP and SCB. It may appear that MIO plays a minor role in the DPPH assay.

3.       The results discussion on the loading and release shown on Fig 11 needs to be expanded from the point of view of drugs that could be suitable or unsuitable for loading/release in the investigated systems. Metronidazole often is used in combination with other drugs. The authors could refer to a recent review on dual and multidrug delivery: Dual and multi-drug delivery nanoparticles towards neuronal survival and synaptic repair, Neural Regen Res. 2017; 12: 886–889. doi: 10.4103/1673-5374.208546

4.       The article does not have in vivo tests from suitable cell lines. Thus the question for the cytotoxicity remains without a clear answer. The authors refer to published results for other MIO based systems, which may not be completely relevant to the present study.

Author Response

Reply to Reviewer #2

Comment-1: The abstract should emphasize what is the novelty of the paper.

Reply-1: Accepted. We have clearly stated the novelty of this manuscript in the abstract and introduction sections.

Comment-2. The results shown on Fig.9 needs to be appended with pure WTP and SCB. It may appear that MIO plays a minor role in the DPPH assay.

Reply-2: Accepted and provided. Yes, we agree with your comment. MIO-NPs do not have a significant role in antioxidant activity (Current line 456-460).

Comment-3. The results discussion on the loading and release shown in Fig 11 needs to be expanded from the point of view of drugs that could be suitable or unsuitable for loading/release in the investigated systems. Metronidazole often is used in combination with other drugs. The authors could refer to a recent review on dual and multidrug delivery: Dual and multi-drug delivery nanoparticles towards neuronal survival and synaptic repair, Neural Regen Res. 2017; 12: 886–889. doi: 10.4103/1673-5374.208546

Reply-3: Thank you so much for your observation. We have read the mentioned article and cited it in our manuscript as it highlights precious information regarding multidrug delivery. The investigation on multi-drug delivery would enhance our manuscript, but this time we only focused only on a single-drug system. Besides, we have added this point in the limitation section (Current line 547-549).

Comment-4. The article does not have in vivo tests from suitable cell lines. Thus the question for the cytotoxicity remains without a clear answer. The authors refer to published results for other MIO based systems, which may not be completely relevant to the present study.

Reply-4: Thank you so much for your scholastic suggestion. Initially, we designed in vitro drug release system and we are planning to work with in vivo experiments for toxicity analysis. We have added this point in the limitation section (Current line 546-547) and omitted the irrelevant articles from the comparative studies part.

Reviewer 3 Report

The article submitted by Naznim et al. reviews deals with the preparation and characterization of magnetic iron oxide incorporated waste tissue paper and sugarcane bagasse nanocomposite particles focused on drug delivery and medical applications. The topic is relevant in the field and brings new input on these nanosystems' characterization regarding antioxidant properties and drug delivery potential. The manuscript is well structured, but there are some questions to be addressed by the authors, which are:

1. How did authors monitor metronidazole stability in media? More data associated with metronidazole quantification is required. For example, which values were obtained for linearity, sensitivity and reproducibility?

2. Every characterization testing made in replicates should discriminate the meaning of "n". Discriminate the difference between n as several batches and n as several sub-batches. The average of 3 different samples of the same batch may be not the same of average of different batches.

3. Since no DLS (Dynamic light scattering) measurements were made, doubts arise about polidispersity and aggregation of nanocomposites. This weakness (?) of the manuscripts requires the authors' deep approach.

4. Releasing data provided by authors (cumulative release of metronidazole over 240 min) is not enough to calculate and conclude the releasing behaviour of nanocomposites. Please provide releasing results over time, and any burst release should be addressed carefully by authors,

Examples of raised issues can be found in the manuscript as follows:

Page 6

Line 201

What does µg stand for? Amount of dried nanoparticles? Mass of nanoparticle-containing dispersion?

Page 7

Line 225

Is metronidazole medium aqueous? pH?

Was metronidazole stability over three days assessed?

Line 229

"after separating the particles from the supernatant." How were particles separated from the supernatant?

Line 230

Was the metronidazole calibration curve validated according to compendia or guidelines?

Page 8

Line 294

"lose matrix and the agglomeration of NCPs was reduced as revealed in SEM image [25]." I cannot follow the authors as there is no decrease in aggregation.

Page 11, Line 3

Does IOM-NPs refer to MIO-NPs. If yes, correct it in the remaining part of the manuscript.

Page 12

Table 1

This table contains too much information and the displayed results can hardly be compared. Information provided by authors in the text is enough.

Page 13

Lines 402-403

Authors refer to nanoparticles' antioxidant properties, not disclosing which type they are referring to.

Page 15

Line 453-454

Disclosed data are not enough to assess releasing behaviour of nanocomposites. Please provide the release of metronidazole over time.

Author Response

Reply to Reviewer #3

Comment-1: The article submitted by Naznim et al. reviews deals with the preparation and characterization of magnetic iron oxide incorporated waste tissue paper and sugarcane bagasse nanocomposite particles focused on drug delivery and medical applications. The topic is relevant in the field and brings new input on these nanosystems' characterization regarding antioxidant properties and drug delivery potential. The manuscript is well structured, but there are some questions to be addressed by the authors, which are:

Reply: Thank you very much for positive response to our manuscript. We have tried our best to revise the overall manuscript and we hope this time the manuscript will be appreciable. Thank you so much for your valuable time and kind consideration.

  1. How did authors monitor metronidazole stability in media? More data associated with metronidazole quantification is required. For example, which values were obtained for linearity, sensitivity, and reproducibility?

Reply-1: Accepted. We prepared a standard calibration curve from a standard stock solution and provided the related data regarding linearity, sensitivity, and reproducibility in the quality control section. Furthermore, we studied intra-day and inter-day precision for stability measurements (Current line 262-306).

Comment-2: Every characterization testing made in replicates should discriminate the meaning of "n". Discriminate the difference between n as several batches and n as several sub-batches. The average of 3 different samples of the same batch may be not the same as the average of different batches.

Reply-2: Accepted and provided the required information (Current line 309-310).

Comment-3: Since no DLS (Dynamic light scattering) measurements were made, doubts arise about polydispersity and aggregation of nanocomposites. This weakness (?) of the manuscripts requires the authors' deep approach.

Reply-3: Thank you for your suggestion. As we did not perform the DLS measurements, we have added this issue in the limitation section (Current line 549-551).

Comment-4: Releasing data provided by authors (cumulative release of metronidazole over 240 min) is not enough to calculate and conclude the releasing behavior of nanocomposites. Please provide releasing results over time, and any burst release should be addressed carefully by the authors.

Reply-4: Thank you so much for your scholastic observation. Analysis of releasing results over time and burst release would provide more information, we took data from 10, 20, 40, 60, 80, 100,120, 140, 160,180, 200, 220 and 240 min respectively but we showed data for 240 min only in our manuscript following the reference. Based on this comment we will design our next experiments.

Examples of raised issues can be found in the manuscript as follows:

Comment-5: Page 6, Line 201: What does µg stand for? Amount of dried nanoparticles? Mass of nanoparticle-containing dispersion?

Reply-5: Accepted and resolved. µg stands for the mass of nanoparticle-containing dispersion (Current line 214-217).

Comment-6: Page 7, Line 225: Is metronidazole medium aqueous? pH? Was metronidazole stability over three days assessed?

Reply-6: Accepted and resolved. Metronidazole was taken in a phosphate buffer solution of pH 7.4. The stability of metronidazole over three days was assessed and the data were provided in the quality control section.

Comment-7: Line 229 "after separating the particles from the supernatant." How were particles separated from the supernatant?

Reply-7: Accepted and resolved. (Current lines 244-245 and 257-258).

Comment-8: Line 230 Was the metronidazole calibration curve validated according to compendia or guidelines?

Reply-8: Accepted and provided. The calibration curve was prepared according to the International Conference On Harmonisation (ICH) guideline (Current line 265-276).

Comment-9: Page 8, Line 294 "lose matrix and the agglomeration of NCPs was reduced as revealed in SEM image [25]." I cannot follow the authors as there is no decrease in aggregation.

Reply-9: Accepted and the sentence was corrected. We found that there is still aggregation present (Current line 355-356).

Comment-10: Page 11, Line 3, Does IOM-NPs refer to MIO-NPs. If yes, correct it in the remaining part of the manuscript.

Reply-10: Yes, it was a typing error. We have corrected it throughout the manuscript.

Comment-10: Page 12, Table 1, This table contains too much information and the displayed results can hardly be compared. Information provided by authors in the text is enough

Reply-10: Accepted. We have omitted this table.

Comment-10: Page 13, Lines 402-403, Authors refer to nanoparticles' antioxidant properties, not disclosing which type they are referring to.

Reply-10: Accepted and provided the required information (Current line 467).

Comment-11: Page 15, Line 453-454, Disclosed data are not enough to assess releasing behaviour of nanocomposites. Please provide the release of metronidazole over time.

Reply-11: Thank you so much for your scholastic observation. We evaluated the drug-releasing efficiency of NCPs and NPs for 240 min and we did not take further data. Based on this comment we will design our next experiments. However, we have added this point in the limitation section (Current line 544-545).

Round 2

Reviewer 1 Report

There are still some problems:

1.    There are a few grammatical mistakes, such as lines 263-264, 276-277, 350, 385, and so on.

2.    The discussion in lines 359-369 didn't mean anything, it should be rewrote.

3.    The number of section was confused, such as “2.8. Quality control”, “2.8. Statistical analyses”, “4. Limitations”, “4. Conclusions”.

4.    “4. Limitations” and “4. Conclusions” should be combined and re-discussed.

5.    In the manuscript, the mechanism of drug-loading and drug-releasing on the cellulose-SCB and SCB/MIO-NCPs or cellulose-WTP and WTP/MIO-NCPs should be carefully discussed in the paper. Moreover, the effect of MIO-NCPs, ZnO and Ag on drug-loading and drug-releasing should be presented.

6.    The English of the manuscript should be improved.

Author Response

Dear Editor-in-Chief,                                                                                2023-02-11

Pharmaceutics, MDPI

Thank you so much for your cooperation regarding our manuscript entitled (manuscript ID: pharmaceutics-2141763) " Synthesis of magnetic iron oxide incorporated cellulose composite particles: An investigation on antioxidant properties and drug delivery applications". We have carefully revised the overall manuscript and addressed all the issues raised by the respected reviewers and the changes were highlighted in yellow color in the revised manuscript. We would like to bestow thanks to the honorable reviewers for their valuable suggestions and constructive comments. All the relevant corrections and answers to the reviewer’s comments were summarized below.

We appreciate very much if you could consider this manuscript for the publication in Pharmaceutics.

Yours sincerely

Hasi Rani Barai, Ph.D.

Professor

Department of Mechanical Engineering

Yeungnam University, Gyeongsan 38541, Korea

E-Mail: [email protected]

Reply to Reviewer #1

Comment-1. There are a few grammatical mistakes, such as lines 263-264, 276-277, 350, 385, and so on.

Reply-1: Accepted. We have corrected those mistakes.

Comment-2. The discussion in lines 359-369 didn't mean anything, it should be rewrote.

Reply-2: Accepted and rewritten those lines (Current line 362-369).

Comment-3. The number of section was confused, such as “2.8. Quality control”, “2.8. Statistical analyses”, “4. Limitations”, “4. Conclusions”.

Reply-3: Thank you so much for your scholastic observation. We have accepted and corrected those typing errors.

Comment-4. “4. Limitations” and “4. Conclusions” should be combined and re-discussed.

Reply-4: Accepted and combined these sections (Current line 553-561)

Comment-5. In the manuscript, the mechanism of drug-loading and drug-releasing on the cellulose-SCB and SCB/MIO-NCPs or cellulose-WTP and WTP/MIO-NCPs should be carefully discussed in the paper. Moreover, the effect of MIO-NCPs, ZnO and Ag on drug-loading and drug-releasing should be presented.

Reply-5: Accepted. We have extended this section 3.2.3. Moreover, the effect of MIO-NCPs, ZnO, and Ag in drug delivery has been added.

Comment-6. The English of the manuscript should be improved.

Reply-6: Accepted. We have tried our best to revise the overall manuscript and fixed all the grammatical errors. Hope this time the manuscript will be appreciable.

Reply to Reviewer #2

Comment-1: The authors improved the paper. From my side it can be accepted.

Reply-1: Thank you so much for accepting our manuscript. Your valuable comments are inspiring to us.

Reviewer 2 Report

The authors improved the paper. From my side it can be accepted.

Author Response

Dear Editor-in-Chief,                                                                     2023-02-11

Pharmaceutics, MDPI

Thank you so much for your cooperation regarding our manuscript entitled (manuscript ID: pharmaceutics-2141763) " Synthesis of magnetic iron oxide incorporated cellulose composite particles: An investigation on antioxidant properties and drug delivery applications". We have carefully revised the overall manuscript and addressed all the issues raised by the respected reviewers and the changes were highlighted in yellow color in the revised manuscript. We would like to bestow thanks to the honorable reviewers for their valuable suggestions and constructive comments. All the relevant corrections and answers to the reviewer’s comments were summarized below.

We appreciate very much if you could consider this manuscript for the publication in Pharmaceutics.

Yours sincerely

Hasi Rani Barai, Ph.D.

Professor

Department of Mechanical Engineering

Yeungnam University, Gyeongsan 38541, Korea

E-Mail: [email protected]

Reply to Reviewer #2

Comment-1: The authors improved the paper. From my side it can be accepted.

Reply-1: Thank you so much for accepting our manuscript. Your valuable comments are inspiring to us.
